# Adaptive Vision Token Selection for Multimodal Inference

## Abstract

Vision encoders typically generate a large number of visual tokens, providing information-rich representations but significantly increasing computational demands. This raises the question of whether all generated tokens are equally valuable or if some of them can be discarded to reduce computational costs without compromising quality. In this paper, we introduce a new method for determining feature utility based on the idea that less valuable features can be reconstructed from more valuable ones. We implement this concept by integrating an autoencoder with a Gumbel-Softmax selection mechanism, that allows identifying and retaining only the most informative visual tokens. Experiments show that the sampler can reduce effective tokens and inference FLOPs by up to **50%** while retaining **99-100%** of the original performance on average. On challenging OCR-centric benchmarks, it also surpasses prior SOTA. The sampler transfers to the video setting as well: despite minor drops, zero-shot results remain strong without video-specific training. Our results highlight a promising direction towards adaptive and efficient multimodal pruning that facilitates scalable and low-overhead inference without compromising performance.

## 1 Introduction

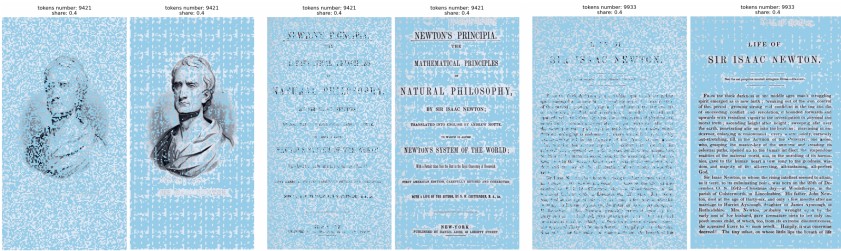

Figure 1: Comparison of feature selection methods on Newton's Principia text, in each pair of images: random feature selection retaining 40% of tokens (left), and our proposed feature selector retaining 40% of tokens (right).

In recent years, vision encoders have become important components for various downstream tasks, providing universal representation of visual features. These encoders are trained to effectively compress raw pixel information into latent embeddings. Depending on their training objectives, vision encoders can encapsulate different types of information in their hidden states. However, it is widely recognized that many of these encoded features contain redundant or irrelevant information for downstream tasks (Raghu et al., 2022; Naseer et al., 2021; Tong et al., 2024). Therefore, reducing the number of output features produced by vision encoders is an important and challenging task — especially now, as encoders increasingly serve as fundamental mechanisms for visual understanding in multimodal models (Li et al., 2024b; Chen et al., 2024c; Tong et al., 2024).

Multimodal models that process visual inputs typically condition on outputs of a Vision Transformer (ViT) (Dosovitskiy et al., 2021), appending a long vision-derived prefix to the input of a Large Language Model (LLM) via a projection layer. Although this method gives promising results, handling large context length (especially when processing high-resolution images) remains a

significant challenge. Moreover, previous studies have observed that not all ViT outputs equally contribute to downstream task performance (Devoto et al., 2024); many tokens can be redundant, noisy, or simply irrelevant (Yang et al., 2024a). Therefore, selectively identifying and retaining only the most informative features can significantly decrease the number of tokens while maintaining model performance.

To address this issue, we propose a novel method to select the most informative visual features from the encoder output using an autoencoder-based approach implemented with Gumbel-Softmax sampling. Our method identifies features that are essential to preserve crucial visual information, allowing us to accurately reconstruct the original feature set. We show that this training procedure not only efficiently identifies valuable features, but also provides interpretable results, highlighting informative features that clearly correspond to specific parts of the image in the pixel space. Furthermore, we illustrate how our approach can be seamlessly integrated during inference in multimodal models, significantly reducing the visual prefix length without compromising performance.

In experiments conducted with the LLaVA-NeXT (Li et al., 2024b) with various large language models (LLMs) backbones and InternVL (Chen et al., 2024c) series of models, we demonstrate that features selected using the the proposed approach contain essential information for the model to provide the correct answer to most of the analyzed tasks. Notably, our method reduces visual context length by up to 50% with minimal performance degradation in most benchmarks. Besides, our method significantly outperforms the current SOTA approaches on the OCR-based tasks, such as document and chart question answering.

The contributions of our paper can be summarized as follows:

- We propose a novel method for selecting the most informative features from vision encoders.

- We demonstrate how our approach serves as an effective in-place feature reduction method for existing multimodal models without requiring further fine-tuning.

- We empirically confirm that retaining as little as 50% of the original visual features can be sufficient to maintain near-baseline performance on multiple multimodal benchmarks.

- Our method outperforms most existing baselines on the complex OCR-based benchmarks.

## 2 RELATED WORK

Recently, several approaches for reducing context in multimodal models have been proposed, operating either at the vision-encoder level or within LLM layers and thus shortening context at different points in the stack.

### 2.1 TOKEN PRUNING

Pruning removes irrelevant/low–value Vision Transformer tokens while retaining salient information. Many methods use attention scores for selection (Tang et al., 2023), and some promote *diversity* to preserve broader coverage (Long et al., 2023). High-quality embeddings are especially critical for detection/segmentation (Liu et al., 2024). Task-specific variants exist; e.g., Kinfu & Vidal (2023) propose three pose-estimation pruners guided by a lightweight pose head or learnable joint tokens. The common goal is to keep tokens most informative for the downstream task.

### 2.2 TOKEN GENERATION AND MERGING

A complementary line compacts representations by generating/merging tokens. Token-Learner (Ryoo et al., 2021) produces a small set of learned tokens; Token Merging (Feng & Zhang, 2023) forms "meta-tokens" by adaptively merging similar ones; (Lee & Hong, 2024) uses learnable decoupled embeddings for end-to-end merging; Resizable-ViT (Zhou & Zhu, 2023) predicts token-length labels to keep informative tokens. Hierarchical backbones such as PVT (Wang et al., 2021) downsample tokens stage-wise (static, content-agnostic), reducing cost for high-res inputs; Li et al. (2023b) further examine tokenization choices.

While effective on classic CV tasks (classification, detection, segmentation), most pruning/merging techniques are tailored to single-modal vision settings and transfer poorly to vision-language models (VLMs), which must preserve visually relevant evidence aligned with text.

## 2.3 Vision Context Reduction in Multimodal Models

Reducing visual context is crucial in multimodal models because visual tokens can dominate the LLM's sequence length; yet, their utility is query dependent. Existing strategies operate at different points in the stack. Interpolation–style methods downsample features while attempting to preserve salient content (e.g., LLaVA-OneVision (Li et al., 2024a)); the InternVL family (Chen et al., 2025) leverages pixel-unshuffle for high-resolution inputs; and trainable token compressors or learnable queries (e.g., Perceiver IO (Jaegle et al., 2022), BLIP-2 (Li et al., 2023c)) are built into the architecture and typically require joint training.

A more plug-and-play line prunes tokens post-hoc. FASTV (Chen et al., 2024a) prunes vision tokens in selected LLM layers using attention scores from earlier layers; PyramidDrop (Xing et al., 2025) ranks image tokens with a lightweight attention module and drops a fixed fraction at multiple depths. Methods such as HiRED (Arif et al., 2024) and VISIONZIP (Yang et al., 2024b) select tokens using attention scores from the [CLS] token of the vision encoder. However, these scores degrade as pruning ratios increase (Guo et al., 2024), leading to underperformance on high-resolution, detail-dense images (e.g., text-heavy/OCR tasks). Diversity-based selectors — DivPrune (Alvar et al., 2025), PACT (Dhouib et al., 2025), and HiPrune (Liu et al., 2025) — promote coverage by clustering or maximizing token diversity and keeping the most representative tokens; some are training-free, others require light finetuning for the best performance.

The proposed method does not depend on the LLM during selection. Instead, it scores tokens by how well they preserve core visual information in the encoder's representation. Because it is training-free relative to the VLM, it can be applied directly in both purely visual (vision encoder level) and multimodal pipelines (as vision context compressor).

## 3 Useful Feature Selection

The Transformer architecture has been successfully used as a backbone for vision encoders (Dosovitskiy et al., 2021), providing hidden representations suitable for a wide range of vision tasks. However, due to the inherent design of the self-attention mechanism in Transformers, neighboring tokens naturally contain information about each other. Consequently, we assume that information may be duplicated redundantly in different regions of the output feature tensor. In particular, some visual representations could potentially be composed entirely of information already present in other tokens. If such redundant representations exist, they can be identified and removed without causing significant performance degradation in vision-related tasks.

This hypothesis naturally raises two critical questions: how can one quantitatively measure whether one set of features contains more information than another, and how can one select the optimal subset of features?

### 3.1 Feature Subset Comparison

For any image $I$, the corresponding feature set $F$ has dimensions $(L, C)$, where $L$ is the number of vision tokens, and $C$ is the corresponding dimension of each vision token. Tokens identified for potential exclusion have the characteristic property that they can be reconstructed from the remaining visual tokens in the set. Thus, if there exists an optimal reconstruction function $R$, which takes a pruned subset of features as input $F^{pr}$ (where the superscript $pr$ denotes *pruned*) with dimensions $(L^{pr}, C)$ and returns a reconstructed set $F^{rec}$ with dimensions $(L, C)$, and if a proximity function $dist$ is defined between two tensors, then one subset is considered superior to the other if it allows for a more accurate reconstruction of the discarded visual tokens.

Formally, subset $F_1^{pr}$ is superior to subset $F_2^{pr}$ if:

$$dist\left(R(F_1^{pr}), F\right) < dist\left(R(F_2^{pr}), F\right). \tag{1}$$

## 3.2 How to Select the Optimal Set?

To select the most informative features, we aim to find a function $S$, referred to as the *optimal selector*, which takes $F$ as input and returns a pruned subset $F^{pr}$. We train this selector in a way similar to the autoencoder:

$$\min_{\theta,\psi} \ \mathrm{dist}\Big( R_\psi(S_\theta(F)), \ F \Big) + L^{pr}. \tag{2}$$

In this formulation, the first term provides a high-quality reconstruction of the original feature set from the pruned subset. The additional term $L^{pr}$ penalizes the selector $S_\theta$ if it trivially selects all tokens (acting as an identity function), thereby encouraging a more concise but informative subset.

## 4 Method

### 4.1 Implementation Details

In this section, we present the implementation details of the approach described in Section 3, which consists of two main components: *Feature Selector* and *Reconstructor*.

#### 4.1.1 Feature Selector Architecture

Feature Selector $S$ consists of four Transformer layers and a Gumbel-Softmax-based (Jang et al., 2017) head. The head creates a binary mask, as shown in Figure 2 (left), where zeros indicate visual tokens to be removed and ones indicate tokens to be retained.

During training, feature embeddings corresponding to zeros in the binary mask are replaced by a shared learnable embedding $E_{masked}$, (this embedding will be reconstructed later by the component described in 4.2). During inference, embeddings corresponding to zeros are simply discarded, while those corresponding to ones are kept for downstream task. For example, they can be used as image representations in Vision-Language models, as shown in our experiments in Section 5.

For more flexibility during inference, one can choose to use logits from the linear layer instead of a hard binary mask. Based on these logits, the user can select a fixed number of the most informative features. This is exactly the approach that is used in our experiments, which we describe in Section 5.

### 4.2 Reconstructor Architecture

The Reconstructor is divided into two parts: a Feature Reconstructor $R^f$ and an Image Reconstructor $R^{im}$.

Feature Reconstructor $R^f$ consists of four Transformer layers and Image Reconstructor $R^{im}$ consists of two Transformer layers and upsampling layers in interleaved with residual blocks to restore the spatial resolution. The primary objective of $R^f$ is to restore the tokens that were replaced by the learned embedding $E_{masked}$, after which $R^{im}$ should recover the image as shown in Figure 2 (left).

### 4.3 Loss Function

As described in Section 3.2, the optimization objective is formulated as the sum of two terms: (1) a reconstruction loss and (2) a regularization term that aims to minimize the amount of information required for reconstruction.

We decompose the reconstruction term into two parts:

$$dist(F^{rec}, F) + dist(I^{rec}, I), \tag{3}$$

where $F^{rec}$ is reconstruction of features and $I^{rec}$ is reconstruction of image.

In principle, we would expect the reconstruction loss to approach zero while the regularization term converges to the fraction of useful visual tokens. However, in practice we did not observe the expected behavior. We found that the optimizer is more likely to converge to a local minimum where

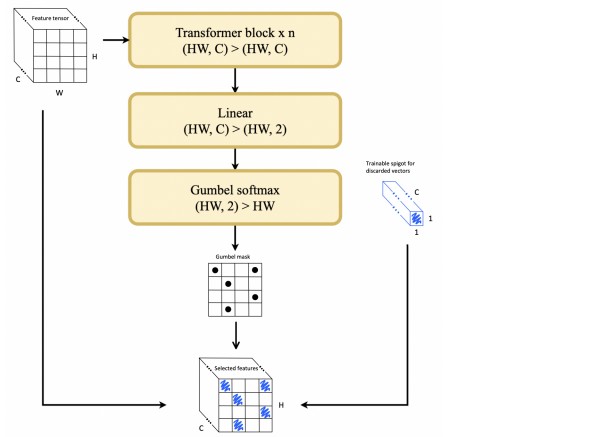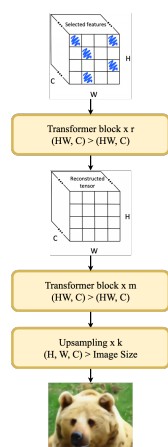

Figure 2: **Left:** Illustration of the Feature Selector in training mode. It uses four Transformer layers and a Gumbel-Softmax head to generate a binary mask where zeros mark tokens for removal and ones for retention. During training, the masked embeddings are replaced by a shared learnable embedding. During inference, the masked embeddings are discarded, while the retained ones are used for downstream tasks, such as image representations in Vision-Language models. **Right:** Illustration of Feature Reconstructor's functionality. Its primary objective is to restore the tokens that were replaced with a learned representation and then reconstruct the full image.

the regularization term drops to zero, thereby avoiding the token utilization penalty. More details can be found in Appendix B.1.

To resolve this issue, we modify the regularization term as follows.

$$L^{pr} \rightarrow \max(L^{pr}, p) - \min(L^{pr}, q), \tag{4}$$

where $p$ and $q$ specify the range of values within which the average proportion of selected tokens should fluctuate.

In other words, whenever $L^{pr}$ falls within the interval $(q, p)$, the regularization penalty is effectively disabled. Empirically, we observe that $L^{pr}$ first decreases to $p$ and then fluctuates around it for the remainder of the training period, while the reconstruction loss continues to decrease.

### 4.4 TRAINING

As shown in Figure 2, our approach is similar to VQ-VAE (van den Oord et al., 2018), except that we use a set of input features instead of a learned dictionary, and the latent representation may vary in size.

We train $R^f_\psi$, $R^{im}_\xi$ and $S_\theta$ following the framework introduced in 3.2. Specifically, we choose the $l_2$ norm for the distance function dist and compute $L^{pr}$ using the mask generated by $S_\theta$.

**Feature Selector.** The feature selector $S_\theta$ processes the original feature tensor $F$ and outputs a subset of selected features $F^{pr}$ along with a binary mask $M$, referred to as the "Gumbel mask" as illustrated in Figure 2 (right). Formally, this can be expressed as:

$$F^{pr}, \ M \ = \ S_\theta(F), \tag{5}$$

where the mask $M$ specifies which spatial locations of the input tensor $F$ are retained (marked as ones) and which are discarded (marked as zeros). The output $F^{pr}$, labeled as "Selected features" in Figure 2 (left), is formed by replacing the discarded feature vectors with a shared learnable representation (shown as blue hatched vectors).

**Feature Reconstructor.** The reconstructor is defined by:

$$F^{rec} \ = \ R^f_\psi(F^{pr}) \ \text{and} \ I^{rec} \ = \ R^{im}_\xi(F^{rec}), \tag{6}$$

with $F^{rec}$ denoting the "Reconstructed tensor" shown in Figure 2 and $I^{rec}$ denoting the image in the same figure.

**Regularization Term.** Regularization term is computed directly from the mask:

$$L^{pr} = \sum_{h=0,w=0}^{H,W} \frac{M_{h,w}}{HW}. \tag{7}$$

**Overall Objective.** Incorporating the modified regularization from 4.3, the overall optimization problem can be defined as follows:

$$\min_{\psi,\xi,\theta} \left( \left\| F^{rec} - F \right\|_2 + \left\| I^{rec} - I \right\|_2 \right) + \alpha_1 \cdot \max\left(L^{pr}, p\right) - \alpha_2 \cdot \min\left(L^{pr}, q\right). \tag{8}$$

In our experiments $p = 0.6$, $q = 0.1$ and $\alpha_1 = \alpha_2 = 0.1$. All components are fully differentiable, and we optimize them using gradient descent.

### 4.5 DATASET

For our training dataset, we sampled 115K images from the COCO dataset (Lin et al., 2015), 9k images from DocVQA train set (Mathew et al., 2021c) and 9k images from ChartQA train set (Masry et al., 2022b). Each image was pre-processed with a specific vision encoder for which the selector was trained. The resulting feature representations were used as training data.

### 4.6 TRAINING HYPERPARAMETERS

During training, we set loss weights to $p$=0.6, $q$=0.1, and $\alpha_1$=$\alpha_2$=0.1. Optimization uses Adam with a cosine schedule and $5\%$ warmup. We use a batch size of 32 and train for 100 epochs in three stages (60+20+20 epochs). Learning rates are $5\times10^{-6}$ for InternVL-like models and $1\times10^{-5}$ for LLaVA-like models. These settings were fixed empirically and not exhaustively tuned.

## 5 EXPERIMENTS

### 5.1 EXPERIMENTAL SETUP

We evaluate our feature selector by integrating it with the vision encoders that serve as backbones in several multimodal model families: LLaVA/LLaVA-NeXT, LLaVA-Video (Zhang et al., 2025) (CLIP-based visual encoder (Radford et al., 2021)), and the InternVL (InternViT encoder (Chen et al., 2024b)) series. The method is plug-and-play: once trained for a given vision encoder, the selector can be attached to a vision — language model without any additional fine-tuning. Furthermore, a selector trained for a specific encoder can be reused across VLMs that employ that encoder, even when the encoder has been further fine-tuned during VLM training.

We test models augmented with our selector under multiple pruning ratios and compare them against the strongest recent pruning methods, including HiPrune (Liu et al., 2025), PACT (Dhouib et al., 2025), PDrop (Xing et al., 2025), FastV (Chen et al., 2024a), and DivPrune (Alvar et al., 2025). As a baseline, we also, evaluated random feature selection at matched ratios across all the tasks, to estimate whether the evaluated tasks suffer from random pruning the multimodal context.

### 5.2 BENCHMARK

We evaluate on a diverse suite of multimodal benchmarks. For general and academic-domain VQA, we use **AI2D** (Hiippala et al., 2020), **MMMU** (Yue et al., 2024), and **ScienceQA** (Lu et al., 2022). For OCR-centric tasks with high-resolution images — where preserving reading-relevant tokens is critical — we include **DocVQA** (Mathew et al., 2021b), **ChartQA** (Masry et al., 2022a), **In-foVQA** (Mathew et al., 2021a), and **TextVQA** (Singh et al., 2019). To assess hallucination sensitivity under pruned context, we additionally report results on **VizWiz** (Gurari et al., 2018) and **POPE** (Li et al., 2023d). For the video evaluation, we used 5 benchmarks ActivityNet-QA (Yu et al., 2019), SeedBench (Li et al., 2023a), NextQA (Xiao et al., 2021), EgoSchema (Mangalam et al., 2023), and LongVideoBench (Wu et al., 2024).

Table 1: Comparison of pruning approaches on Open-LLaVA-Next (Vicuna-7B). All methods are training-free. **Bold** denotes the best result with a margin of $\geq 2$ percentage points (pp) over the runner-up; underline denotes results within $< 2$ pp of the best.

| Model | DocVQA | ChartQA | InfoVQA | TextVQA | SQA img | VizWiz | MMMU | Avg |
|---|---|---|---|---|---|---|---|---|
| *2880 Tokens (100%, $\approx$ 21 TFLOPs)* | | | | | | | | |
| Original | 70.5 | 64.1 | 33.1 | 67.3 | 70.4 | 61.8 | 38.1 | 100% |
| *160 Tokens (5.6%, $\approx$ 1 TFLOP)* | | | | | | | | |
| HiPrune | 20.8 | **32.7** | 20.1 | 40.6 | 67.7 | 54.6 | 36.9 | 64.3% |
| **Ours** | **32.0** | 27.4 | 20.5 | **54.4** | 68.0 | 54.8 | 36.0 | **69.4%** |
| *320 Tokens (11.1%, $\approx$ 2 TFLOPs)* | | | | | | | | |
| DivPrune | 39.7 | 37.9 | 22.1 | 59.6 | 67.4 | 58.7 | **38.1** | 76.9% |
| HiPrune | 41.2 | **42.2** | 23.4 | 55.8 | **68.2** | 58.8 | 37.7 | 78.3% |
| **Ours** | **46.6** | **42.2** | 24.1 | **61.2** | **68.2** | 57.1 | 35.0 | **80.8%** |
| *640 Tokens (22.2%, $\approx$ 4 TFLOPs)* | | | | | | | | |
| DivPrune | 50.7 | 46.6 | 23.7 | 61.7 | 68.5 | 59.5 | 37.1 | 83.6% |
| HiPrune | 58.3 | 47.7 | 26.8 | 60.8 | 69.4 | 60.4 | 38.2 | 87.5% |
| **Ours** | **61.9** | **57.8** | 28.1 | **65.7** | 69.0 | 59.4 | 37.0 | **92.4%** |
| *1152 Tokens (40%, $\approx$ 8 TFLOPs)* | | | | | | | | |
| DivPrune | 58.1 | 51.0 | 25.8 | 63.3 | 68.9 | 60.0 | 36.7 | 88.2% |
| HiPrune | 65.6 | 51.6 | 28.7 | 62.3 | 69.4 | 61.1 | 37.3 | 91.7% |
| PDrop | 64.4 | 58.6 | 32.2 | 65.7 | 69.4 | 61.6 | 38.1 | 95.5% |
| **Ours** | **68.3** | **63.3** | 31.0 | **67.2** | 69.9 | 60.7 | 37.4 | **97.8%** |
| *1440 Tokens (50%, $\approx$ 10 TFLOPs)* | | | | | | | | |
| DivPrune | 62.4 | 52.6 | 27.0 | 64.4 | 68.9 | 59.8 | 37.1 | 90.4% |
| HiPrune | 67.2 | 53.0 | 29.0 | 62.4 | 69.2 | 61.2 | 37.3 | 92.6% |
| **Ours** | **69.6** | **64.2** | **32.3** | **67.2** | 70.5 | 60.7 | 37.1 | **99.1%** |

## 5.3 FLOPs Calculation

Following (Chen et al., 2024a; Xing et al., 2025), we count only FLOPs attributable to *vision tokens*, including multi-head attention (MHA) and feed-forward (FFN). Let $n$ be the number of vision tokens (e.g., 2880 for LLaVA-Next with $1+4$ crops), $d$ the hidden size (e.g., 4096), $m$ the FFN intermediate size (e.g., 11008), and $l$ the number of LLM layers (e.g., 32). The LLM contribution over a fraction $\alpha n$ tokens is

$$\text{FLOPs}_{\text{LLM}} = \left(4(\alpha n)d^2 + 2(\alpha n)^2 d + 2(\alpha n)dm\right) l.$$

Our sampler adds a *per-crop* stage over $n_c$ tokens for each of $C = n/n_c$ crops. Using the same MHA+FFN form, with $l_s$ sampler layers (we use $l_s = 4$),

$$\text{FLOPs}_{\text{sampler}} = \frac{n}{n_c}\left(4n_c d^2 + 2n_c^2 d + 2n_c dm\right) l_s.$$

Thus, the reported total is $\text{FLOPs}_{\text{total}} = \text{FLOPs}_{\text{LLM}} + \text{FLOPs}_{\text{sampler}}$.

## 5.4 Experimental Results

The proposed sampler-based pruning method outperforms the current SOTA overall and on OCR-centric tasks. Table 1 reports results on OPEN-LLAVA-NEXT (VICUNA-7B) (Chen & Xing, 2024). Across moderate-high pruning ratios (11-50%), our sampler consistently surpasses prior methods on **DocVQA**, **ChartQA**, and **TextVQA**; even at the most aggressive 5.6% setting, it still leads on DocVQA and TextVQA. With 50% of tokens retained, it essentially matches the unpruned model on OCR-based benchmarks.

Considering sampler overhead, the inference cost drops from $\approx$21 TFLOPs to $\approx$10 TFLOPs while retaining **99.1%** of the average performance.

Table 2: Comparison of pruning approaches on LLaVA-1.6 (Mistral-7B).

| Model | DocVQA | ChartQA | InfoVQA | SQA img | MME | MMMU | AI2D | Avg |
|---|---|---|---|---|---|---|---|---|
| *2880 Tokens (100%)* | | | | | | | | |
| Original | 63.6 | 52.9 | 30.5 | 72.6 | 317.5 / 1504.4 | 35.2 | 67.4 | 100% |
| *864 Tokens (30%)* | | | | | | | | |
| PACT | 54.9 | 44.5 | 28.5 | 72.9 | 312.1 / 1484.9 | 34.8 | 67.0 | 95.2% |
| Ours | **59.8** | **50.8** | 28.4 | 73.4 | **327.1 / 1507.6** | **37.3** | 66.6 | **99.3%** |
| *1296 Tokens (45%)* | | | | | | | | |
| PACT | 62.5 | 51.7 | 30.0 | 73.2 | 307.1 / 1504.0 | 34.7 | 67.1 | 99.0% |
| Ours | 62.8 | 53.1 | 29.6 | 73.1 | **321.8 / 1516.3** | 35.6 | 66.9 | 100% |

Table 2 illustrates comparison of our method with the PACT model that was applied for LLaVA-1.6 (Mistral-7B) model. At matched token budgets on LLaVA-1.6, our sampler outperforms PACT across key tasks. At 30% tokens, it improves DocVQA, ChartQA, MMMU, MME, and SQA-img and reaches **99.3%** Avg (vs. 95.2% for PACT). At 45% tokens, it again leads on most benchmarks, achieving 100% Avg (on par with the unpruned model) and even slightly exceeding the original on several tasks.

### 5.4.1 VIDEO EXPERIMENTS

Table 3 reports video-benchmark results with several pruning methods on LLAVA-NEXT-VIDEO-7B. Although our sampler is not trained specifically for video, it transfers well, delivering strong zero-shot performance and trailing DIVPRUNE only slightly. We anticipate further improvements with video-specific fine-tuning.

Table 3: Results on video benchmarks for LLaVA-Video-7B. All experiments use up to 8 frames per video.

| Model | TFLOPs | ActivityNet | SeedBench | NextQA | EgoSch | LongVideo Bench |
|---|---|---|---|---|---|---|
| Original | 7.8 | 2.49 / 49.7 | 43.6 | 27.2 | 40.3 | 43.5 |
| FastV | 1.1 | 1.95 / 33.9 | 33.0 | 22.5 | 29.1 | – |
| DivPrune | 1.1 | 2.45 / 48.5 | 41.07 | 26.1 | 40.0 | 42.4 |
| Sampler (our, $\alpha$ =15%) | 1.2 | 2.28 / 45.6 | 42.1 | 25.9 | 35.0 | 40.7 |
| Sampler (our, $\alpha$ =10%) | 0.8 | 2.23 / 44.7 | 40.8 | 25.5 | 34.3 | 40.1 |

### 5.5 ABLATION STUDY

**Random-token pruning.** As a control, we randomly mask a fixed fraction of vision tokens at each compression rate and evaluate performance, comparing against our sampling-based selector. We further demonstrate portability beyond LLaVA by training the sampler for INTERNVL3 with an INTERNVIT encoder. Figure 3 summarizes INTERNVL3 results across compression rates.

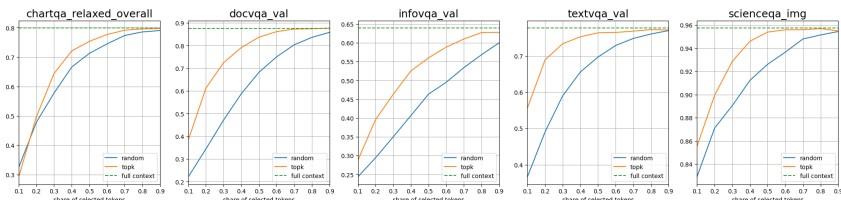

Figure 3: Comparison of InternVL3 performance across several benchmarks with various compression rate.

Across all compression rates, our sampler markedly outperforms the random baseline, indicating that these benchmarks are sensitive to vision — context pruning and that our method effectively

removes irrelevant tokens from the visual input. Qualitative examples of the sampler are provided in Appendix A (Figure 6).

**Sampler architecture and training epochs.**    We ablate the sampler on InternVL, varying encoder depth (1-8 layers) and training budget. Across the sweep, the 4-layer Transformer encoder consistently performs best, and a 100-epoch budget is required to reach the strongest results on smaller fraction of the remaining tokens. Figures 4 and 5 demonstrate INTERNVL3 performance as a function of sampler depth and training epochs, respectively.

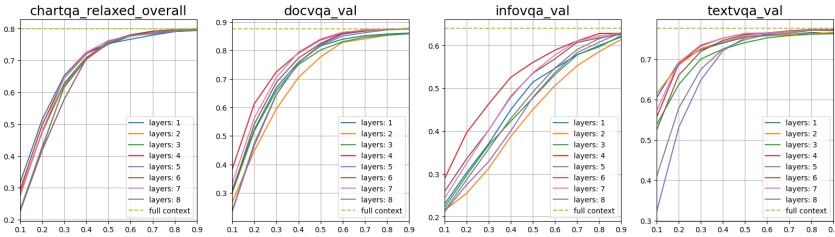

Figure 4: Sampler architecture ablation study for InternVL3 model.

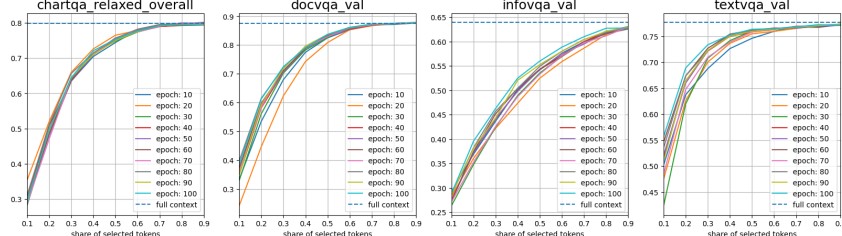

Figure 5: Training epochs ablation study for InternVL3 model.

## 6    CONCLUSION

We introduced a sampler-based method for selecting informative features from visual encoders. The sampler is implemented as a trainable VAE with a Gumbel–Softmax bottleneck integrated into a ViT, and it reduces the number of output vision tokens while preserving the most salient ones. We applied the method in a plug-and-play manner to modern VLMs (Open-LLaVA-Next with multiple LLM backbones and InternVL). Experiments show that the sampler can reduce effective tokens and inference FLOPs by up to **50%** while retaining **99-100%** of the original performance on average. On challenging OCR-centric benchmarks, it also surpasses prior SOTA. The sampler transfers to the video setting as well: despite minor drops, zero-shot results remain strong without video-specific training.

However, we acknowledge certain limitations of the proposed method. For long videos, compression without text conditioning can underperform. In future work, we plan joint fine-tuning of the selector and the language model, and a study of hybrid strategies that combine interpolation-style compression with Gumbel-based selection to improve compatibility and robustness.

Overall, these results point to promising directions for extracting compact, informative visual representations, enabling faster inference, lower memory footprint, and improved resilience to noisy visual inputs.

## REPRODUCIBILITY STATEMENT

Section 4.4 describes the sampler's training setup, covering datasets and hyperparameters. Reproducibility code is provided in the supplementary materials. Please refer to the README file in the supplementary materials to configure the environment and reproduce the sampler training.

## ETHICS STATEMENT

This work introduces a method for reducing inference compute in multimodal models and does not create new artifacts (e.g., datasets or benchmarks). All experiments use publicly available datasets and open-source models widely adopted by the community.

In this work, LLM was used exclusively for editorial refinement. It did not affect the study design, data, analysis, or outcomes.

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

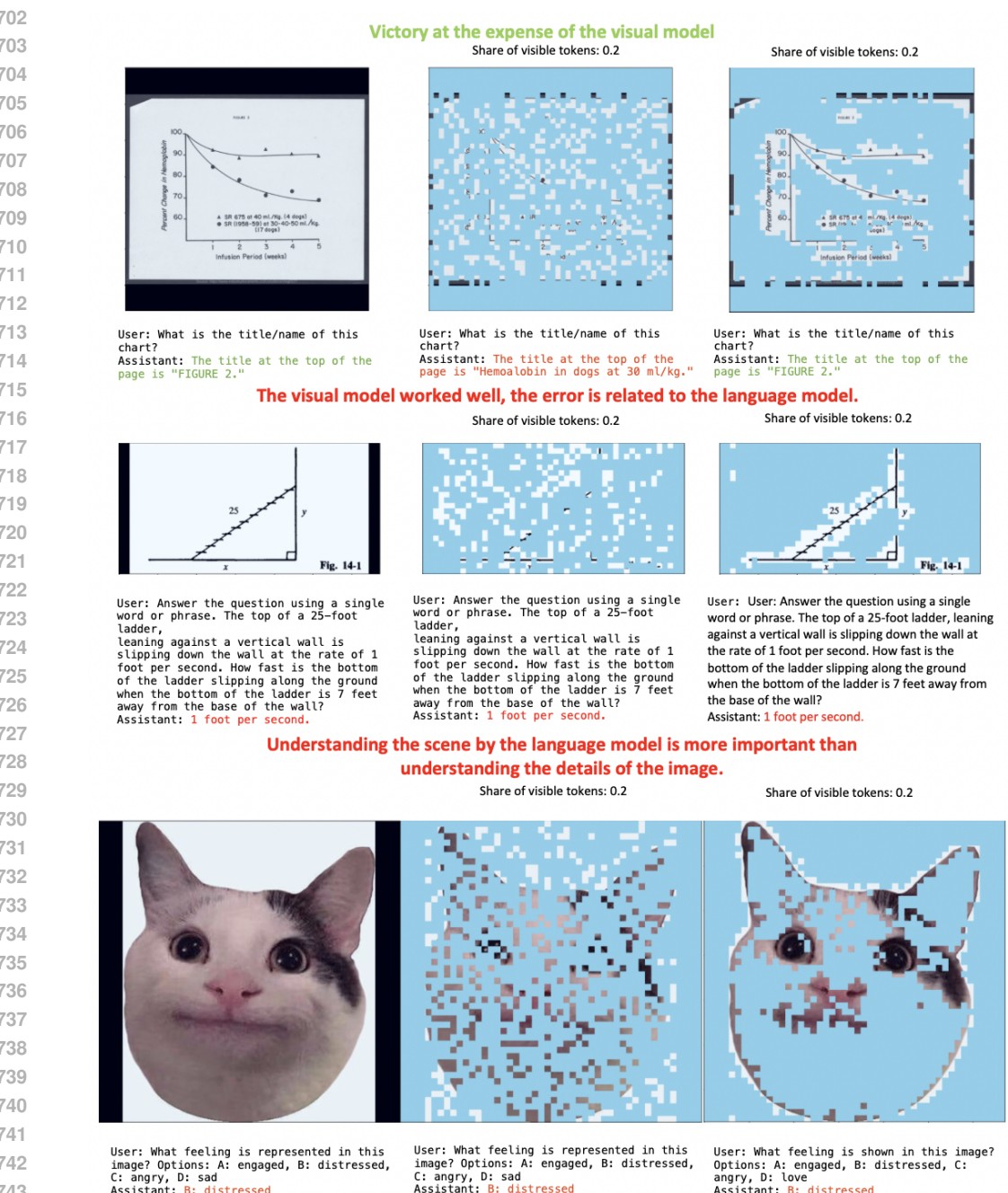

Figure 6: Images from three benchmarks illustrating cases where the vision-language model gives correct answers or makes errors. The first column shows the model's responses using the full visual context, the second column uses a randomly selected set of features, and the third column uses the features selected by our selector. (1) DocVQA: to answer the question selecting the correct features is crucial. (2) MMMU (math): to answer this question, both visual understanding and logical reasoning are important, but the model fails to reason correctly. (3) MMstar: the image details are less important, and the language model plays a dominant role.

# A QUALITATIVE EXAMPLES

## A.1 SELECTOR ON VARIOUS FEATURE PERCENTAGE

Figure 6 illustrates the effect of keeping only 20% of vision tokens. From left to right: original image, tokens kept by random selection, and tokens kept by our sampler. Compared to random, the sampler focuses on regions that contain the most informative content, which in turn improves the model's answer to the final question.

## A.2 FAILED EXAMPLES

In this section, we evaluate failure cases in which the VLM's answers degrade when the feature selector is applied. To investigate the causes of such failures, we analyzed detailed logs of the model's predictions on the benchmarks, focusing specifically on cases where the initial model answer was correct but, after applying the selector, the model changed its answer to an incorrect one. Based on the observed performance degradation at specific feature levels, we propose the following hypotheses:

- The selector did not retain specific image features that are crucial for answering the question.
- The model itself could not adapt to the modified distribution of image features and failed to answer correctly even when the mask contained the necessary information.

Our analysis shows that most failures correspond to the second hypothesis. In other words, in most cases all the information necessary to answer the question was present in the image, however, the model could not successfully leverage it and changed its answer. Please refer to Figure 7 for more details.

We hypothesize that this behavior is caused by the specifics of the model's training: changing the standard number of vision tokens to a sparser set may lead to performance degradation. In future work, we will study this effect in more detail by performing ablations and fine-tuning the model on the reduced feature space, then comparing performance.

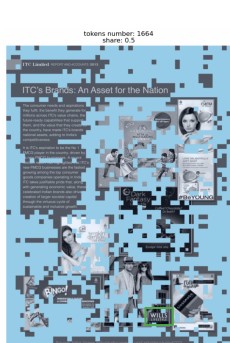 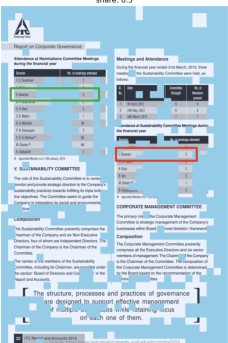 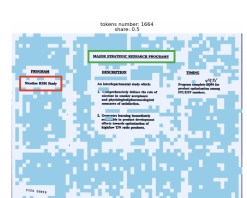 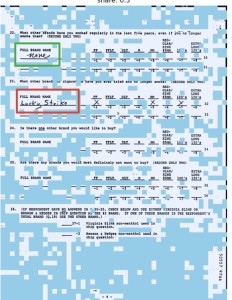

(a) **Question:** What is the name of the fashion wear/clothing advertise?
**Target:** Wills Lifestyle
**Answer:** WILLS

(b) **Question:** How many nomination committee meetings has S. Banerjee attended?
**Target:** 2
**Answer:** 3

(c) **Question**: What is the name of the research program?
**Target**: MAJOR STRATEGIC RESEARCH PROGRAMS
**Answer**: nicotine rsm study

(d) **Question**: What is the brand name for the 22nd question?
**Target**: none
**Answer**: lucky strike

Figure 7: Examples of model errors. Green boxes indicate areas containing the correct answer, red boxes indicate areas that the model likely focused on when providing the answer.

We observe that the information necessary to provide the correct answer remains visible after applying the mask; however, the model is no longer able to answer correctly, even though its answer on the full image was correct.

# B ABLATION STUDY

## B.1 GUMBEL USAGE ABLATION

Equation 8 presents the loss function used to optimize the network. In Section 4.3, we explained that the term $\alpha_1 \cdot L^{pr}$ is replaced by the term $\alpha_1 \cdot max(L^{pr}, p) - \alpha_2 \cdot \min(L^{pr}, q)$. Otherwise, we observed that the optimizer falls into a local minimum of the term $L^{pr}$, i.e., it drives $L^{pr} = 0$.

It turns out that this behavior of the optimizer is related to the use of Gumbel Softmax for computing the Gumbel mask (the mask of sampled tokens). When using a simple straight-through estimator $(y_{hard} - y_{soft}.detach() + y_{soft}$, where $y_{soft}$ is the Softmax), the problem of getting stuck in a local minimum is not observed, meaning the standard term $\alpha_1 \cdot L^{pr}$ can be used in the loss function.

We compared three versions for selector training objectives:

- (Our) Gumbel Softmax + $\left( \left\| F^{rec} - F \right\|_2 + \left\| I^{rec} - I \right\|_2 \right) + \alpha_1 \cdot \max(L^{pr}, p) - \alpha_2 \cdot \min(L^{pr}, q)$.

- Simple straight-through estimator + $\left( \left\| F^{rec} - F \right\|_2 + \left\| I^{rec} - I \right\|_2 \right) + \alpha_1 \cdot L^{pr}$.

- Simple straight-through estimator + $\left( \left\| F^{rec} - F \right\|_2 + \left\| I^{rec} - I \right\|_2 \right) + \alpha_1 \cdot \max(L^{pr}, p) - \alpha_2 \cdot \min(L^{pr}, q)$.

Figure 8 shows the benchmark performance of the InternVL3 model on four representative benchmarks, with the selectors fine-tuned using different methods. Apart from the differences described above, the training pipelines are identical in all experiments $\alpha_1 = \alpha_2 = 0.1$. We observe that higher performance at lower token percentages is obtained by training with Gumbel addition, thus, the randomness introduced by Gumbel addition acts as a form of regularization.

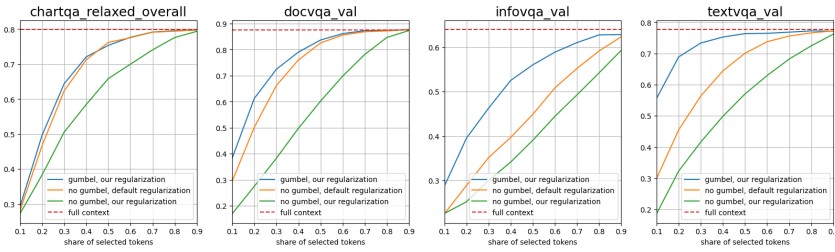

Figure 8: A comparison of training objective: (*blue*) Gumbel Softmax with the proposed loss regularization; (*orange*) straight-through estimator with the standard $L_{pr}$ term; (*green*) straight-through estimator with the proposed loss regularization.

## B.2 INSTABILITY OF "GUMBEL SOFTMAX + LOSS WITH A STANDARD REGULARIZATION TERM" APPROACH

In some of our experiments, we attempted to train a feature selector using a Gumbel Softmax + loss with a standard regularization term: $\left( \left\| F^{rec} - F \right\|_2 + \left\| I^{rec} - I \right\|_2 \right) + \alpha_1 \cdot L^{pr}$. However, this method proved to be very sensitive to training hyperparameters. An example of training graphs in such a pipeline is shown in Figure 9. We believe that there exists an optimal subset of hyperparameters for which such training would be stable; however, searching for it is quite resource-consuming, so we instead chose to improve the regularization term so that regularization is disabled in the case of collapse described in Equation 4.

## B.3 THE STABILITY OF OUR APPROACH

As shown in Figure 9 in the previous section, using a regularization term of the form $\alpha_1 \cdot L^{pr}$ leads to training instability, manifested by the regularization term collapsing to zero. Figure 10 shows the

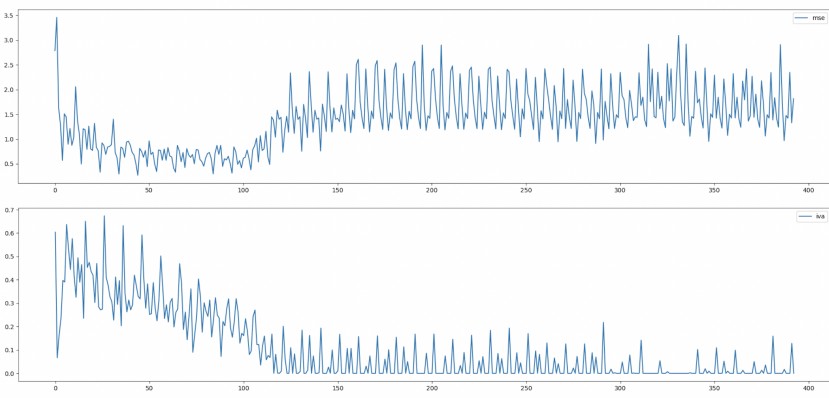

Figure 9: Selector training curves for the InternVL model with the Gumbel Softmax pipeline + loss with standard regularization. The curve on top corresponds to term $\left\| F^{rec} - F \right\|_2 + \left\| I^{rec} - I \right\|_2$. The curve on the bottom corresponds to term $\alpha_1 \cdot L^{pr}$.

optimization results with our regularization term $\alpha_1 \cdot \max\left( L^{pr}, p \right) - \alpha_2 \cdot \min\left( L^{pr}, q \right)$. As can be seen, the nature of the dynamics changes dramatically: from a certain point, the regularization term begins to fluctuate around the value $p$. This prevents training degradation and ensures a more robust reconstruction loss optimization process.

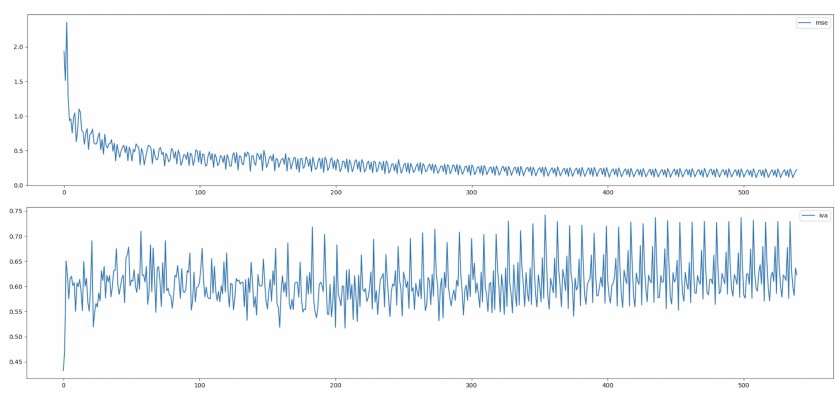

Figure 10: Selector training curves for the InternVL model with the Gumbel Softmax pipeline + loss with our regularization term. The graph on top corresponds to term $\left\| F^{rec} - F \right\|_2 + \left\| I^{rec} - I \right\|_2$. The curve on the bottom corresponds to our regularization term $\alpha_1 \cdot \max\left( L^{pr}, p \right) - \alpha_2 \cdot \min\left( L^{pr}, q \right)$.

### B.4 CONDITIONAL SELECTION

The proposed approach operates at the vision-encoder level, allowing us to reduce the number of vision tokens while preserving the most essential information about the image. One promising direction for future work is to condition the feature selector on the text query. To achieve this, we change the learning and insight extraction strategy. Specifically, instead of feeding the selector with features from the visual encoder, we provide it with features from a certain LLM layer, which are already enriched with textual information. Examples of such conditioned sampling are shown in Figure 11.

We found that our approach works when placing the question before the image. However, reversing the image and question significantly impacts the baseline model's metrics. We also found that the metrics drop less when placing the question both before and after the image.

Table 4: Comparison of the original InternVL3-2B model and the same model with a conditional selection of features.

| Model | DocVQA | ChartQA | InfoVQA | TextVQA | MME | GQA | POPE | SQA img | VizWiz |
|---|---|---|---|---|---|---|---|---|---|
| Original | 82.3 | 71.8 | 52.8 | 73.7 | 413.2 / 1320.2 | 54.0 | 89.5 | 81.6 | 60.7 |
| w/ Condition (middle layer) | 83.1 | 72.4 | 53.1 | 73.9 | 427.9 / 1300.7 | 54.7 | 89.0 | 81.8 | 60.8 |

The Table 4 shows the results of a comparison between the original model and the same one with middle-layer conditioning (that is, only question-relevant tokens remain after the middle layer). In both cases, the question is presented as $\langle question \rangle + \langle image \rangle + \langle question \rangle$.

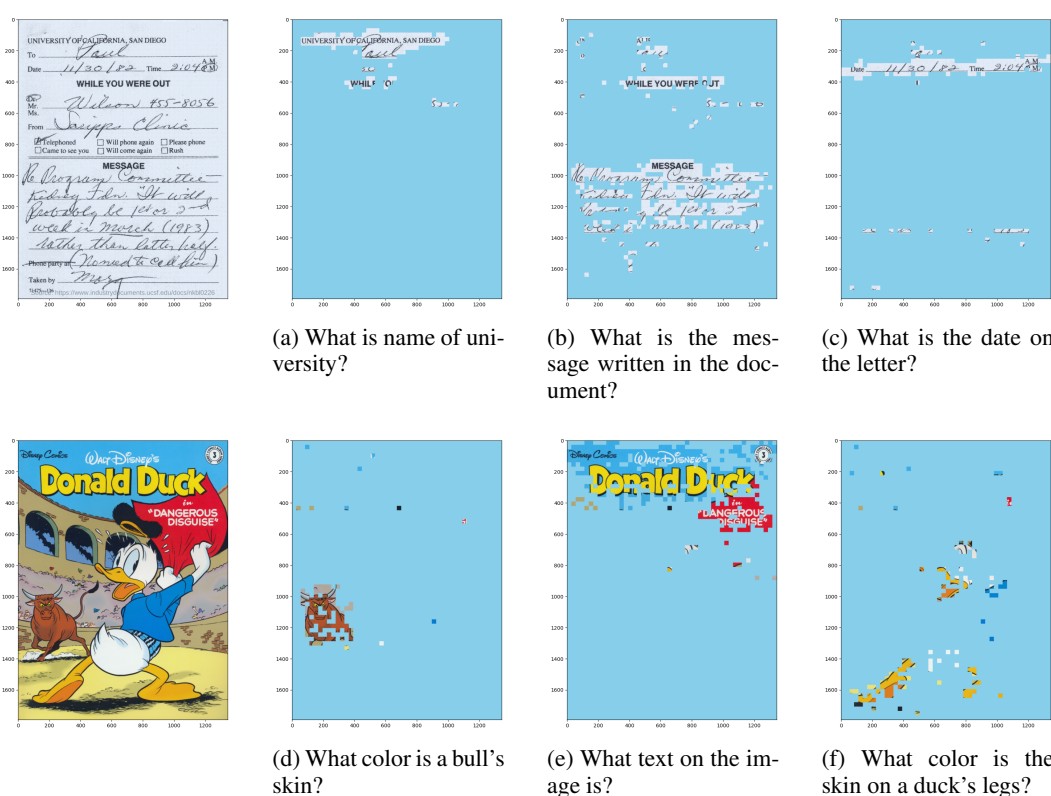

(a) What is name of university?

(b) What is the message written in the document?

(c) What is the date on the letter?

(d) What color is a bull's skin?

(e) What text on the image is?

(f) What color is the skin on a duck's legs?

Figure 11: Examples of conditional selectors in action. Each example is accompanied by the question for which the mask was generated.

## B.5 STABILITY OF THE SELECTION MASK

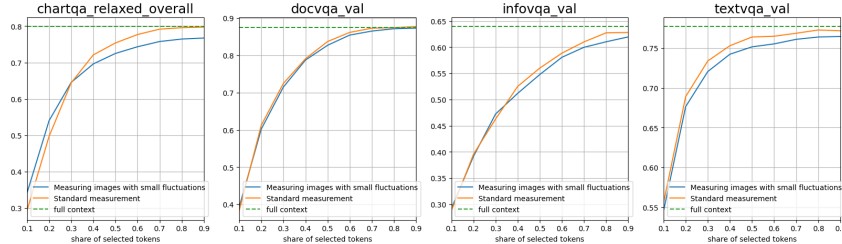

Figure 12: Stability of the selection mask ablation study for InternVL3 model.

To test the mask's robustness to small distortions, we applied two transformations to the images. First, we cropped up to five percent of the image on each side independently (the crop percentage

was randomly selected). Second, we varied the brightness with a random coefficient in the range of 0.8-1.2. The measurement results are shown in the Figure 12.

