# OpenReview forum: "Adaptive Vision Token Selection for Multimodal Inference"
_ICLR.cc/2026/Conference — Submitted to ICLR 2026_

### Official Review · Reviewer_t3o2 · 2025-10-28

**Soundness:** 1
**Presentation:** 2
**Contribution:** 2
**Rating:** 2
**Confidence:** 4

**Summary:**

The paper presents a token pruning method for inference on images in which a subset of image embedding tokens are selected for the purpose of downstream tasks. The method involves a subset selector and a reconstructor, trained in the style of VQVAE. Some improvements over prior token pruning baselines are presented.

**Strengths:**

1. The problem statement - can we determine the optimal subset for reconstructing the image and replace the existing tokens? - is interesting.
2. Some improvements compared to recent token pruning methods like DivPrune, HiPrune, PDrop.

**Weaknesses:**

1. Unfortunately, I believe the training methodology is seriously flawed. I'll explain in bullet points:
    - The Gumbel softmax trick is for **sampling** from a categorical distribution in a differentiable way. Since the mask creation (as described) is deterministic, you don't need this, a simple straight-through estimator (y_hard - y_soft.detach() + y_soft where y_soft is the softmax) would suffice.
    - In fact adding Gumbel noise like this may potentially harm the training, as there is no need of sampling the tokens for masking unless intended for some regularization purposes (like dropout)
    - Even ignoring this, **selecting the indices** from the straight through estimator defeats the whole point. The gradients cannot propagate through a selection operation, this is the reason you find that just having L_pr as regularizer drives it down to zero, as there is no optimization pressure from the reconstruction loss.
    - After your modification to the L_pr, the method somewhat works because this essential turns off the regularizer once it is in a certain range between p to q. This is similar to simply having the token selector output a fixed number of tokens and throwing out the regularizer, it is not a joint optimization at all.
    - The correct way of doing this would be something like M * E_img + ( 1- M ) * E_masked, the gradients can propagate through this operation just fine.

    I hope the authors can fix these serious issues in future versions of the paper. My suggestions: (1) replace Gumbel-softmax with just a straight-through estimator, and (2) use linear weighting instead of index selection.

2. Other weaknesses are present (token merging baselines are absent, improvements over existing baselines seem marginal) but are minor compared to the serious issues in the training.

**Questions:**

Please address the weaknesses listed above

---

> ### Author Response · Authors · 2025-11-14
> **Clarification on Gumbel Noise and Regularization**
>
> Dear reviewer,
>
> Thank you very much for your thoughtful comments on our paper. Below, we have addressed some of your concerns. We hope that, guided by your feedback, we will be able to improve our work. We kindly ask you to elaborate as thoroughly as possible on the range of issues you see in the current version of the paper, so that we can use your insights to further enhance it.
>
> ---
>
> **Reviewer comment:**
> *The Gumbel softmax trick is for sampling from a categorical distribution in a differentiable way. Since the mask creation (as described) is deterministic, you don't need this; a simple straight-through estimator (`y_hard - y_soft.detach() + y_soft`, where `y_soft` is the softmax) would suffice. In fact, adding Gumbel noise like this may potentially harm the training, as there is no need to sample the tokens for masking unless intended for some regularization purposes (like dropout).*
>
> Indeed, we did not test whether the method works without the randomness introduced by the Gumbel noise as a form of regularization. We will run such an experiment and report the results to you.
>
> ---

---

> > ### Author Response · Authors · 2025-11-14
> > **Gradient Flow and Differentiability in Masking Implementation**
> >
> > **Reviewer comment:**
> > *Even ignoring this, selecting the indices from the straight-through estimator defeats the whole point. The gradients cannot propagate through a selection operation. This is the reason you find that just having `L_pr` as regularizer drives it down to zero, as there is no optimization pressure from the reconstruction loss. After your modification to the `L_pr`, the method somewhat works because this essentially turns off the regularizer once it is in a certain range between `p` and `q`. This is similar to simply having the token selector output a fixed number of tokens and throwing out the regularizer—it is not a joint optimization at all. The correct way of doing this would be something like `M * E_img + (1 - M) * E_masked`; the gradients can propagate through this operation just fine.*
> >
> > You mention that the training code contains operations (index selection) that prevent gradients from flowing. Let’s walk through the code (from the supplementary materials) together and clarify how it works, in order to resolve this misunderstanding.
> >
> > The training code for the selector (e.g., for the `InternVL3` model) is located in the directory `supplementary/source_code/InternVL3/code/train` and is executed via the `train.py` file. In the `train` function (line 179), a standard training loop is performed, which includes the following lines (191–193):
> >
> > ```python
> > 191 out, out_fe, input, check_dict = model(fe, im)
> > 192 loss = model.loss_function(im, fe, out, out_fe, check_dict['input_vector_amount'])
> > 193 loss['loss'].backward()
> > ```
> >
> > Let us examine the model’s `forward` pass and the loss computation in detail. We begin with the `forward` method.
> >
> > To do this, we refer to the `model.py` file, where the `forward` function is defined on line 563. Within this function, line 567 is likely the source of concern, as it may involve non-differentiable operations. (We note upfront that all operations before line 567 and all operations after it are clearly differentiable—they consist either of Transformer layer calls, residual block invocations, `.view()` operations, or slicing such as `[:, 1:]` used to remove the CLS token.)
> >
> > ```python
> > 567 quantized_latents, val_dict = self.vq_layer(encoding, features)
> > ```
> >
> > We are interested in two questions:
> > 1. Do gradients flow through `quantized_latents`?
> > 2. Do gradients flow through `val_dict["input_vector_amount"]`?

---

> ### Author Response · Authors · 2025-11-14
> **Gradient Flow and Differentiability in Masking Implementation 2**
>
> The first matters because we reconstruct the masked features and the original image from `quantized_latents`. The second is important because it serves as the regularization term in the loss function.
>
> Here, `self.vq_layer` is an instance of the `VectorQuantizer` class (defined on line 411). Therefore, calling `self.vq_layer(...)` invokes the `forward` method of that class.
>
> Let us carefully step through the code of this function and trace the variables `quantized_latents` and `val_dict["input_vector_amount"]`:
>
> ```python
> 424   def forward(self, latents, input): # [B, L, D], [B, L, D]
> 425       logits = self.linear(latents) # [B, L, 2]
> 426
> 427       one_hotes = gumbel_softmax(logits, hard=True) # [B, L, 2]
> 428       forma_batch = torch.zeros_like(latents) # [B, L, D]
> 429       forma_batch += self.forma[None, None, :] # [B, L, D]
> 430
> 431       quantized_latents = input * one_hotes[:, :, :1] + forma_batch * one_hotes[:, :, 1:] # [B, L, D]
> 432       input_vector_amount = torch.mean(one_hotes[:, :, :1]) # [1]
> 433
> 434       inds = torch.arange(one_hotes.shape[1], device=one_hotes.device)[None].repeat(latents.shape[0], 1)
> 435       gumbel_mask = one_hotes[:, :, 0] > 0.5
> 436       num_items = gumbel_mask.sum(dim=1)
> 437       encoding_inds = inds[gumbel_mask]
> 438       encoding_inds = encoding_inds.split(num_items.tolist())
> 439
> 440       val_dict = {
> 441           'input_vector_amount': input_vector_amount,
> 442           'encoding_inds': encoding_inds,
> 443           'input_norm': input.norm(dim=-1),
> 444       }
> 445       return quantized_latents, val_dict
> ```
>
> As can be seen in line 431:
> ```python
> quantized_latents = input * one_hotes[:, :, :1] + forma_batch * one_hotes[:, :, 1:]  # [B, L, D]
> ```
> the computation of `quantized_latents` exactly matches the formula you proposed:
> $E_{\text{img}} \cdot M + E_{\text{masked}} \cdot (1 - M)$.
>
> Here:
> - `input` == $E_{\text{img}}$,
> - `forma_batch` == $E_{\text{masked}}$,
> - `one_hotes[:, :, :1]` == $M$,
> - `one_hotes[:, :, 1:]` == $(1 - M)$.
>
> Therefore, there are no issues with gradient flow at this point.
>
> As can be seen in line 432:
> ```python
> input_vector_amount = torch.mean(one_hotes[:, :, :1])
> ```
> Gradients also flow through this term.
>
> In summary, our training code already implements the exact formulation you suggested. Any confusion likely arose from the wording in the paper. If so, please let us know which part was unclear—we will certainly revise it.
>
> ---
>
> Now let us examine the loss computation function (line 583). Within it, we need to trace the variable `loss`, as it is through this variable that the optimization process occurs. (All other variables in the output were included solely for debugging and monitoring during training.)
>
> ```python
> 583   def loss_function(self, input, input_features,
> 584                           rec, rec_features, input_vector_amount):
> 585       """
> 586       :param args:
> 587       :param kwargs:
> 588       :return:
> 589       """
> 590       rec_loss = F.mse_loss(rec, input)
> 591       rec_features_loss = F.mse_loss(rec_features, input_features)
> 592       iva_factor = torch.tensor(self.iva_factor, device=input_vector_amount.device)
> 593       low_iva_factor = torch.tensor(self.low_iva_factor, device=input_vector_amount.device)
> 594       # loss = rec_loss + rec_features_loss + torch.max(iva_factor, input_vector_amount)
> 595       loss = (rec_loss + rec_features_loss) + \
> 596           0.1 * torch.max(iva_factor, input_vector_amount) - \
> 597           0.1 * torch.min(low_iva_factor, input_vector_amount)
> 598
> 599       normed_recons = rec / torch.norm(rec, dim=1)[:, None]
> 600       normed_input = input / torch.norm(input, dim=1)[:, None]
> 601       latent_cross_product = torch.einsum('bdij,bdij->bij', normed_recons, normed_input) # [B x 14 x 14]
> 602       cos_loss = latent_cross_product.mean()
> 603
> 604       return {
> 605           'loss': loss,
> 606           'msei': rec_loss,
> 607           'msef': rec_features_loss,
> 608           'cos': cos_loss,
> 609           'iva': input_vector_amount
> 610       }
> ```
>
> Thus, the variable `loss` is computed according to the following formula:
> ```python
> 595       loss = (rec_loss + rec_features_loss) + \
> 596           0.1 * torch.max(iva_factor, input_vector_amount) - \
> 597           0.1 * torch.min(low_iva_factor, input_vector_amount)
> ```
>
> As established above, gradients flow correctly through the model’s forward pass. Consequently, the reconstruction term exerts consistent optimization pressure, and the regularization term also contributes gradient signal—except when the input falls within the interval $(q = low\\_iva\\_factor, p = iva\\_factor)$.
>
> In summary, outside the interval $(q, p)$ we obtain joint optimization of both the reconstruction and regularization terms, while inside the interval we optimize only the reconstruction term.

---

> ### Comment · Reviewer_t3o2 · 2025-11-14
> **Thanks for the clarification**
>
> Thank you for the clarification. It seems that the place in the code where you do the selecting is not used for the loss. In that case, the gradients should indeed flow properly.
>
> However, it remains that the simple "reconstruction + alpha x penalty" loss should work if everything was fine during training, and the fact that you need to significantly weaken the penalty via clipping it means that the gradients are not getting computed correctly during training. I suspect it may be that the gumbel noise is diluting the gradient signal (it's not needed here, just use straight through). Otherwise, the issue is more subtle. You will know you fixed the issue once "reconstruction + alpha x penalty" works for some suitable alpha parameter.

---

> > ### Author Response · Authors · 2025-11-21
> > **Response to Reviewer t3o2**
> >
> > Thank you for your comment and idea. At your request, we trained the model in the pipeline you described above. We removed the gumbel softmax, replacing it with a simple straight-through estimator, and also removed the mechanism for disabling the regularization term, replacing it with a simple $\alpha \cdot penalty$ term.
> >
> > We found that the model can be trained within this pipeline. We conducted ablations on several benchmarks on the InternVL model and compared the results between our training method and the method you suggested. Additionally, we also trained the model in a pipeline where we removed the gumbel softmax, replacing it with a simple straight-through estimator, but retained our mechanism for disabling regularization.
> >
> > It turned out that regularization using Gumbel significantly improves training results. We have updated the paper and presented these results in the Appendix, in Chapter B.1: GUMBEL USAGE ABLATION (p.16).
> >
> > So, regularization via Gumbel proves useful, which is why the trick of disabling regularization is necessary, as training in the Gumbel softmax + loss pipeline of the type "reconstruction + $\alpha * penalty$" turns out to be unstable. To help you understand the problem we address with this trick, we've included a training graph in Appendix B in Chapter B.2, "INSTABILITY OF GUMBEL SOFTMAX + LOSS WITH A STANDARD REGULARIZATION TERM." (p.17)

---

> > > ### Comment · Reviewer_t3o2 · 2025-11-25
> > >
> > > Firstly, I thank the authors for taking my feedback into account and conducting the additional insightful experiments. It is great to see that the training converges with just straight-through + default sparsity penalty.
> > >
> > > However, I still feel that the trick of adding Gumbel noise and clipping the sparsity penalty is highly unmotivated and perplexing. The use of Gumbel noise for regularization is completely new. I simply hypothesised that may be a potential reason for keeping it, but this was not mentioned at all in the original paper and the authors seem to have introduced this noise for totally different, unjustified, reasons. Clipping the penalty also feels quite ad-hoc. Both these tricks need to be thoroughly validated before being accepted. I suspect that with the correct $\alpha$, the default formulation should be as good as the one proposed in the work.

---

### Official Review · Reviewer_jQAv · 2025-10-31

**Soundness:** 2
**Presentation:** 3
**Contribution:** 2
**Rating:** 4
**Confidence:** 5

**Summary:**

This paper proposes a trainable selector that keeps only the most informative vision tokens before feeding them to a multimodal model. The selector is a small Transformer with a Gumbel Softmax head that produces a binary mask over encoder features. A reconstructor network is trained jointly so that discarded tokens can be reconstructed from the kept ones, and the objective includes a modified regularizer that nudges the average keep ratio into a desired range p to q. The selector is trained once per vision encoder on a mixture of COCO, DocVQA, and ChartQA features, then attached at inference time to several VLMs without further fine tuning. Experiments claim up to fifty percent token reduction with roughly ninety nine to one hundred percent of baseline accuracy on average, strong gains on OCR tasks, and some transfer to video. FLOPs accounting includes both the LLM and the selector overhead.

**Strengths:**

1. The paper formalizes selection as preserving reconstructability of discarded tokens and implements it with a compact selector plus reconstructor. The approach is conceptually simple and naturally plug and play for many VLMs.
2. The p and q trick for the keep ratio is a concrete engineering fix to a mode collapse problem in the regularizer and is explained with training behavior observations.
3. The selector is attached to Open LLaVA Next and LLaVA 1.6, compared to multiple pruning baselines, and also tried on video models with eight frame inputs.
4. The paper spells out a FLOPs that adds selector cost to LLM cost rather than hiding it, which is important when the selector itself is nontrivial.

**Weaknesses:**

1.  The method keeps tokens that best reconstruct the image features and pixels, not tokens that are most useful to answer a question. This risks throwing away small but question critical text spans or regions that are hard to reconstruct yet crucial for reasoning. The paper acknowledges strong OCR results in some cases, but the core objective remains task agnostic and could be misaligned for many VQA settings.
2. The selector is trained only on COCO plus two OCR datasets, then applied widely across benchmarks and model families. There is no analysis of distribution shift or per benchmark sensitivity to this training mix.
3. The tables emphasize per benchmark accuracy and a FLOPs estimate. There is no thorough report of wall clock latency, throughput at batch sizes relevant to serving, or memory footprint measured on hardware. Without that, it is hard to validate the real system gain when the selector itself adds four Transformer layers per crop.
4. The paper claims ninety nine to one hundred percent of average performance at fifty percent tokens and shows an example figure, but there is little analysis of worst case degradations, error types, or hallucination changes beyond a couple of benchmarks. Averages can hide brittle behavior on rare but safety relevant inputs.
5. The selector is not trained for temporal data and results trail a simple diversity baseline on multiple metrics. The paper positions this as promising transfer, but the drop is material and there is no study of temporal coherence or frame wise instability under pruning.

**Questions:**

1. Can you add a text conditioned variant of the selector that uses the question to guide token importance and compare it with the purely reconstructive objective, especially on question sensitive OCR tasks
2. What are the worst ten percent relative drops across all datasets for a fifty percent keep ratio, and what qualitative failure modes do you observe when the selector deletes rare but crucial regions
3. Please report end to end latency and throughput on a single modern GPU across a realistic prompt distribution, including the selector time and memory, and compare to baselines that are training free and selector free
4. How stable is the mask across seeds and small image perturbations such as slight crops or contrast changes, and is there any evidence of mask flicker across video frames
5. Since the selector is trained on features from a specific encoder, how robust is it to encoders that were further finetuned inside a VLM and to different pooling or crop strategies

---

> ### Author Response · Authors · 2025-11-21
> **Response to Reviewer jQAv**
>
> Thank you for reviewing of our work and valuable comments.
>
> We would like to address some of your questions below.
>
> **1. Can you add a text conditioned variant of the selector that uses the question to guide token importance and compare it with the purely reconstructive objective, especially on question sensitive OCR tasks**
>
> Text-conditioned sampling is a natural extension for our research, and we already performed preliminary experiments on condition-based selection of the input tokens. Let us briefly describe the experimental setup for this.
>
> The proposed approach operates at the vision-encoder level, allowing us to reduce the number of vision tokens while preserving the most essential information about the image. One promising direction for future work is to condition the feature selector on the text query. To achieve this, we modified the training pipeline and insight extraction strategy. Specifically, instead of feeding the selector with features from the visual encoder, we provide it with features from a certain LLM layer, which are already enriched with textual information. We train the model in end-to-end manner with VLM unfreezing. Examples of such conditioned sampling are shown in Appendix B.4 CONDITIONAL SELECTION, Figure 11 (p.18).
>
> We will address the rest of your questions in a few days, once the additional experiments are completed.

---

> > ### Comment · Reviewer_jQAv · 2025-11-26
> > **Response to the authors**
> >
> > I thank the authors for their response. However, most of my main concerns remain insufficiently addressed.
> >
> > * Regarding the text-conditioned selector (Q1), the rebuttal mainly restates that conditional sampling is a natural extension and refers to Appendix B.4, but it does **not** provide a clear, quantitative comparison between the purely reconstructive objective and the text-conditioned variant, especially on question-sensitive OCR tasks as requested.
> > * The rebuttal does **not** answer my question about the **worst 10% relative drops** across datasets at a 50% keep ratio, nor does it provide a qualitative analysis of failure modes when rare but crucial regions are removed (Q2).
> > * There is still **no end-to-end system evaluation**: latency, throughput, and memory under realistic serving conditions, including the selector overhead and a comparison against training-free / selector-free baselines (Q3).
> > * The stability of the selection mask across seeds and small image perturbations (crops, contrast changes), and potential temporal flicker in video, are not discussed or quantified (Q4).
> > * Finally, the robustness of a selector trained on a specific encoder to encoders that have been further finetuned inside a VLM, or to different pooling / cropping strategies, is not addressed (Q5).
> >
> > Given these remaining gaps, my overall assessment is unchanged and I keep my original score and concerns. I encourage the authors, regardless of the final decision, to carefully revisit the questions raised in my original review (in particular Q2–Q5), as addressing them would substantially strengthen the work in a future revision or journal version.

---

> > > ### Comment · Reviewer_jQAv · 2025-11-26
> > > **Response to the authors**
> > >
> > > I thank the authors for their response and for the additional material provided in the appendix. I appreciate the effort to extend the work, especially the exploration of conditional selection and the qualitative analyses. That said, several of my main concerns remain only partially addressed.
> > >
> > > Regarding the text-conditioned selector (Q1), the rebuttal and Appendix B.4 emphasize that conditional sampling is a natural extension and provide illustrative examples in Figure 11. While these examples are helpful and encouraging, there is still no clear quantitative comparison between the purely reconstructive objective and the text-conditioned variant, in particular on question-sensitive OCR benchmarks, which was the core of my original request.
> > >
> > > For Q2, the newly added qualitative failure cases in Appendix A.2 are a welcome addition and help to shed light on some typical errors. However, my question also aimed at a more systematic view of robustness, for instance, reporting the worst 10% relative drops across datasets at a 50% keep ratio and characterizing the corresponding failure modes, especially when rare but crucial regions are pruned. This more global perspective is still missing.
> > >
> > > Concerning Q3, the paper now includes FLOPs-based analysis, which is useful for understanding computational cost at a high level. Nonetheless, an end-to-end system evaluation in terms of wall-clock latency, throughput, and memory on actual hardware, explicitly including the selector overhead and comparing against training-free and selector-free baselines, would be very valuable to substantiate the practical gains in realistic serving conditions.
> > >
> > > For Q4, the stability of the learned masks is not yet evaluated in the way I had in mind. The ablations in Appendix B focus primarily on training behavior and regularization dynamics rather than on robustness of the masks themselves. An analysis of variability across random seeds and sensitivity to small image perturbations (for example, crops and contrast changes) would significantly strengthen the understanding of the method’s reliability.
> > >
> > > Finally, for Q5, I appreciate the clarification that a selector trained for a given encoder can be reused across multiple VLMs that share that encoder, and the empirical results on several models are encouraging. Still, a more explicit study of robustness to encoders that have been further fine-tuned in different VLMs, or to alternative pooling and cropping strategies, would make this claim more concrete and convincing.

---

> > > > ### Comment · Reviewer_jQAv · 2025-11-26
> > > > **response to the authors**
> > > >
> > > > The proposed direction appears promising, and I encourage the authors to carefully address the remaining points mentioned above.

---

> > > > ### Author Response · Authors · 2025-11-28
> > > >
> > > > 1. *Regarding the text-conditioned selector (Q1), the rebuttal and Appendix B.4 emphasize that conditional sampling is a natural extension and provide illustrative examples in Figure 11. While these examples are helpful and encouraging, there is still no clear quantitative comparison between the purely reconstructive objective and the text-conditioned variant, in particular on question-sensitive OCR benchmarks, which was the core of my original request.*
> > > >
> > > > At your request, we have added a table with measurements on various benchmarks to Appendix B.4.
> > > >
> > > > 2. *For Q2, the newly added qualitative failure cases in Appendix A.2 are a welcome addition and help to shed light on some typical errors. However, my question also aimed at a more systematic view of robustness, for instance, reporting the worst 10% relative drops across datasets at a 50% keep ratio and characterizing the corresponding failure modes, especially when rare but crucial regions are pruned. This more global perspective is still missing.*
> > > >
> > > > Sorry, we don't understand what you mean. Most benchmarks assign a deterministic value (either 0 or 1) to the answer to a question, so it's unclear what the worst 10% relative drops are you referring to. Could you please explain in more detail what exactly you want us to measure?
> > > >
> > > > 4. *For Q4, the stability of the learned masks is not yet evaluated in the way I had in mind. The ablations in Appendix B focus primarily on training behavior and regularization dynamics rather than on robustness of the masks themselves. An analysis of variability across random seeds and sensitivity to small image perturbations (for example, crops and contrast changes) would significantly strengthen the understanding of the method’s reliability.*
> > > >
> > > > Thank you for this comment. We conducted a corresponding experiment and presented it in Appendix B.5. We showed that benchmark metrics do not change significantly with small image edge cropping and small brightness changes. Furthermore, our method does not rely on randomness during inference, so it is seed-independent. Since all video frames are processed independently, and small fluctuations do not affect mask quality, the flicker-free video is also evident.

---

> ### Author Response · Authors · 2025-12-03
> **Official Comment by Authors**
>
> We evaluated the average latency of our model with sampling against the original model and the DivPrune method with 50% feature sampling. Our sampling method demonstrates competitive performance across all metrics. While the DivPrune-based approach is faster than the original sampler-free method, it incurs additional overhead for feature selection calculations to determine which features should be retained. Nevertheless, both sampling approaches significantly outperform the original VLM (LLaVA-1.6-Vicuna-7B) in terms of latency. All experiments were conducted on an 1 NVIDIA A100 GPU (80GB) using a single image input, with 3 warmup steps followed by 10 evaluation runs.
>
> | Method | Token Selection | LLM Generate | End-to-End |
> |--------|----------------|--------------|------------|
> | Sampler (0.5) | 6.04 ms | 107.23 ms | 146.84 ms |
> | DivPrune (0.5 | 17.30 ms | 106.54 ms | 157.51 ms |
> | Original | -- | 188.36 ms | 222.12 ms |

---

### Official Review · Reviewer_ChdV · 2025-11-01

**Soundness:** 2
**Presentation:** 2
**Contribution:** 2
**Rating:** 4
**Confidence:** 3

**Summary:**

This paper introduces an adaptive method for selecting visual tokens in multimodal models to address the inefficiency of processing all tokens generated by ViTs. The approach is based on the premise that less informative features can be reconstructed from more valuable ones. An autoencoder with a Gumbel-Softmax selector is employed to identify and retain only the most critical tokens. This method reduces computational cost (FLOPs) by approximately 50% while maintaining 99-100% of the original model performance on downstream tasks. It is designed as a plug-and-play module compatible with existing VLMs without requiring retraining and demonstrates modest generalization to video tasks.

**Strengths:**

1. It achieves high efficiency by reducing computational cost by 50% with negligible loss, a meaningful gain for large-scale inference.
2. Its plug-and-play design requires no retraining of the main VLM, which greatly increases its practicality.
3. The method receives strong empirical validation from comprehensive benchmarks on vision and multimodal datasets, with ablations proving its generality.
4. It demonstrates broad applicability by working across LLaVA, InternVL, and video variants, a rare feat in pruning research.

**Weaknesses:**

1. The theoretical foundation is weak, as the relation between reconstructability and informativeness is intuitive but unproven.
2. Comparisons are limited to some pruning baselines, omitting more visual token selection methods.
3. Claims of video generalization are weakly supported, as the results were only tested on a limited settings of model and benchmark, and no temporal analysis was conducted.
4. Insufficient display and analysis of failed cases.

**Questions:**

1. What is the method's sensitivity to its hyperparameters (p, q, α1, α2), and were any adaptive or learned strategies explored for them?
2. Regarding the video experiments, how is the issue of temporal redundancy addressed? Is token selection performed independently for each frame?
3. Why not conduct experiments and tests on SOTA VLMs like Qwen2.5-VL?
4. Why not present and analyze some failed cases?

---

> ### Author Response · Authors · 2025-11-21
> **Response to Reviewer ChdV**
>
> Thank you for reviewing our work and you valuable comments. Below, we addressed some of your questions, we hope, we could cover them, and if you have additional requests and concerns we will be happy to further discuss them.
>
> **1. What is the method's sensitivity to its hyperparameters (p, q, α1, α2), and were any adaptive or learned strategies explored for them?**
>
> In practice we fixed p=0.6 and q=0.1 across all experiments after initial runs showed these values yielded stable optimization and were rational due to the fact that they show a typical percentage of the tokens essential to create reconstruction. We did not conduct a full sweep over p, q due to compute constraints.
>
> We set $\alpha_1=\alpha_2=0.1$ to provide an additional but non‑dominant signal relative to the reconstruction losses. We are now performing an ablation with $\alpha_1=\alpha_2=0.01$ (10 times lower), the regularizer became too weak: the selector tended toward near‑dense behavior, prioritizing reconstruction and largely ignoring the prior term.
>
> We have also initiated a 10 times higher $\alpha$ term ($\alpha_1=\alpha_2=1.0$). Early training behavior indicates stronger selection pressure with the expected reconstruction-regularization trade‑off. (We will report final metrics once training completes.) We did not find it necessary to decouple $\alpha_1$ and $\alpha_2​$ in the main experiments, but asymmetric slopes are a straightforward extension if one wishes to penalize over p and q in a different manner.
>
> We didn’t learn any of these parameters in the experiments, because they are treated as simple regularization terms that work as intended.
>
> You can see the typical loss curve with the introduced regularization terms in Appendix B, B.3, Figure 10, p.17.
>
> **2. Regarding the video experiments, how is the issue of temporal redundancy addressed? Is token selection performed independently for each frame?**
>
> In this work we did not introduce any temporal dependence across frames, either during training or inference. Our selector is trained on images only, and the video experiments are performed for a zero-shot evaluation of whether the method can transfer to video.
>
> For videos, we apply the token selection to each frame independently, without any mechanism to explicitly handle temporal redundancy between frames. Despite this straightforward per-frame strategy, we obtain results that are on par with the existing baselines. Extending the training and architecture to explicitly model temporal sequences and exploit temporal redundancy in videos is our direction for future work.
>
> **3. Why not conduct experiments and tests on SOTA VLMs like Qwen2.5-VL?**
>
> Thank you for the question. Our choice of baselines was driven by the models on which alternative pruning methods have already been evaluated, so that we could ensure a fair and direct comparison. For this reason, the main model families we focused on in comparison are LLaVA-1.5, LLaVA-Next, and LLaVA-Video.
>
> At the same time, we agree that it is important to test robustness across architectures. Therefore, we additionally evaluated the InternVL-3 family, which adopts a different architectural design and uses pixel shuffle as a context-reduction technique. InternVL-3 is trained on both images and videos and achieves high, often state-of-the-art performance on multimodal benchmarks. In our work, we trained on the InternViT encoder derived from InternVL-3 and reported results to demonstrate applicability beyond LLaVA-based models (please, refer to Section 5.5  for InternVL-3 results). Evaluating our method on more VLMs such as Qwen-VL series, e.g., is a natural extension, and we plan to extend the selector training for them as well.
>
> **4. Why not present and analyze some failed cases?**
>
> In Appendix A.2 (Failed Examples) (please, refer to the revised version of the paper for more details), we present several cases where the model initially produced correct answers when given the full feature set, but failed after feature pruning. We analyzed the top portion of such cases and, interestingly, found that most errors are not due to the sampler, but rather to the model’s behavior on the reduced feature subset. In these examples, the information necessary to answer the question is preserved after applying the selector; however, the model is not able to successfully exploit it. We hypothesize that this effect may be related to the training-free nature of our selector integration: the VLM is adapted to the original distribution and density of image features, and when this distribution becomes sparser, it can sometimes fail to use the remaining features effectively. Additional fine-tuning on sparse feature inputs after selector application may further improve the model's performance.

---

### Official Review · Reviewer_RUPi · 2025-11-02

**Soundness:** 3
**Presentation:** 3
**Contribution:** 2
**Rating:** 4
**Confidence:** 4

**Summary:**

This paper proposes an adaptive visual token selection method for MLLMs. The authors employ a Transformer-based architecture as both the feature selector and the feature reconstructor, and use a Gumbel-Softmax layer to generate the final token masks. The proposed method achieves up to a 50% reduction in FLOPs while maintaining 99–100% of the original performance.

**Strengths:**

1. The authors utilize a VQ-VAE–like method to train an adaptive token selector, while using the model to fit the reconstruction function R and the selector function S, which is a reasonable design.
2. The results shown in Fig. 1 are impressive, as most detailed structures, such as characters, are well preserved, demonstrating the method’s superior capability in high-resolution OCR tasks.
3. The paper is well written and easy to follow, with clear mathematical formulations that guide the reader through the methodology.

**Weaknesses:**

1. The experimental section is somewhat confusing. In Table 1, DivProne is missing at 1T FLOPs, while PDrop is only reported at 8T FLOPs, and neither method appears in Table 2. It would be better to include all methods for a fair comparison.
2. As the authors stated in Section 4.4, the method is similar to VQ-VAE. However, for such a relatively simple approach, more analysis would be valuable, such as explaining why a stacked Transformer block was chosen as both the selector and the reconstructor, and why Gumbel-Softmax was used as the final layer.

**Questions:**

1. In Fig. 4, it would be helpful to include an ablation study using more than four Transformer layers to evaluate the effect of model depth.
2. The choice of Gumbel-Softmax requires further discussion; perhaps a simple linear layer could achieve similar results.
3. Why would this method benefit high-resolution OCR tasks? A theoretical analysis or possible explanation would make this claim more convincing.

**Details Of Ethics Concerns:**

None.

---

> ### Author Response · Authors · 2025-11-21
> **Response to Reviewer RUPi**
>
> Dear reviewer, thank you for your attention to our work and valuable comments. In response to your comments, we have prepared clarifications on several points. We would be grateful if you could point out any other potential shortcomings in our article. This will help us significantly improve its quality.
>
> Below, we address your questions.
>
> **1. In Fig. 4, it would be helpful to include an ablation study using more than four Transformer layers to evaluate the effect of model depth.**
>
> Thank you for your suggestion. We've taken your comment into account and added ablation results of increasing the number of layers in Transformer up to 5, 6, 7, and 8. The results show that the 4-layer model is currently optimal. We've revised the paper text; by downloading the updated version, you'll find additional results in Section 5.5, Figure 4 (p.9).
>
> **2. The choice of Gumbel-Softmax requires further discussion; perhaps a simple linear layer could achieve similar results.**
>
> Based on your comment and the comment of reviewer t3o2, we conducted the experiment described in Appendix B,  B.1 GUMBEL USAGE ABLATION (p.16). We trained a simple straight-through estimator instead of Gumbel, and compared the final performance of the model. It turned out that regularization using Gumbel significantly improves training results.
>
> Also, in the context of this issue, you might be interested in our discussion with the reviewer t3o2.

---

### Author Response · Authors · 2025-12-03
**Official Summary Comment by the Authors**

We would like to thank all reviewers for their valuable feedback, which has helped improve the paper. We would like to summarize the raised questions and undressed concerns below.

1. Transformer architecture ablation (reviewer RUPi). We added a depth ablation and show that a 4-layer transformer selector provides the best trade-off between accuracy and complexity (Section 5.5, Appendix).

2. Gumbel-Softmax ablation (reviewers RUPi, t3o2). As suggested, we implemented a straight-through linear variant without Gumbel-Softmax. It is trainable but underperforms our method, and requires extensive hyperparameter tuning. In contrast, Gumbel-Softmax improves training stability and final metrics while avoiding costly hyperparameter search (Appendix B.1, p.16).

3. OCR performance investigation (reviewer RUPi). Our sampler selects informative regions and reduces background while keeping all selected embeddings (no merging), so information is preserved. We note OCR-specific improvements as promising future work.

4. Reviewer ChdV raised questions about hyperparameter sensitivity and temporal redundancy. We added an ablation for their choice (Appendix B.3, Figure 10, p.17), confirming that they act as regularization terms and are not learned adaptively.

5. Temporal-based training for videos (reviewer ChdV). Our selector is trained on images only; video results are zero-shot transfer, and explicit temporal modeling is left for future work.

6. Failure cases analysis (reviewers ChdV, jQAv). We expanded failure-case analysis (Appendix A.2) and find that most errors stem from the VLM's difficulty using sparse features under training-free integration rather than from information loss.

7. Robustness of sampler through various pooling strategies (reviewer jQAv). We further evaluated our method on strong and diverse VLMs implementing various initial token reduction techniques, including LLaVA-1.5 (no initial context reduction), LLaVA-Next (interpolation), LLaVA-Video, and InternVL-3 (pixel shuffle). The proposed method remains effective across these designs (Section 5.5).

8. We also report latency for our sampler (reviewer jQAv), the sampler-free divprune baseline, and the original LLaVA-1.6-Vicuna-7B model, showing that the sampler adds negligible overhead.

9. Reviewer jQAv also asked about condition-based token selection. We added text-conditioned experiments (Appendix B.4, Figure 11, p.18), demonstrating that the selector can operate on LLM-enriched features and support text-conditioned selection.

10. Reviewer t3o2 requested additional training ablations and raised concerns about baselines. We ran experiments under alternative training regimes and confirm the applicability of our approach (discussed above).

11. Another concern was connected with comparison with merging-based methods. While recent strongest baselines are pruning-based, we also compare to a merging method, PACT, and outperform it (Table 2).

Thank you again for your questions and reviewing our paper.

---

### Meta-Review · Area_Chair_AuUy · 2026-01-07

**Summary:**

This paper proposes an adaptive vision token selection method for multimodal models, where a trainable selector and reconstructor are used to identify and retain informative visual tokens based on reconstructability, aiming to reduce inference cost while preserving performance. The method is designed as a plug-and-play module and is evaluated across multiple VLMs and benchmarks, including OCR-centric and video tasks.

Reviewers raised several substantial concerns that remained largely unresolved. These include questions about `the core objective's alignment with downstream reasoning tasks`, `insufficient robustness and failure-mode analysis`, `lack of realistic system-level evaluation`, and `limited evidence supporting the method's generalization`, particularly to question-critical tokens and video settings. While the empirical results are promising, these issues significantly affect confidence in the method’s reliability and practical impact.

**Reviewer Concerns:**

**1. [Outstanding] Misalignment between reconstructability-based selection and task relevance
(by: `jQAv`, `ChdV`)**

Reviewer `jQAv` raised a central concern that the proposed selector optimizes for reconstructability of visual features, rather than task relevance for downstream VQA or reasoning. This risks pruning small but question-critical regions (e.g., rare text spans in OCR or localized evidence), even if average performance remains high. Reviewer `ChdV` echoed this concern, noting the lack of justification that reconstructability correlates with semantic importance across tasks.

The rebuttal acknowledged this limitation and presented qualitative examples and preliminary text-conditioned variants. However, no systematic quantitative comparison was provided between reconstructive and task-conditioned objectives, especially on question-sensitive benchmarks. As such, the core concern about objective misalignment remains unresolved.

**2. [Partially addressed] Insufficient robustness and worst-case performance analysis
(by: `jQAv`, `ChdV`)**

Reviewers expressed concern that the evaluation focuses on average accuracy, which may mask brittle behavior. Reviewer `jQAv` explicitly requested analysis of worst-case or tail performance and characterization of failure modes when rare but crucial regions are pruned. Reviewer `ChdV` similarly noted the lack of systematic failure analysis.

While the rebuttal added qualitative failure cases in the appendix, it did not provide a global robustness analysis or quantitative tail-risk evaluation. This concern therefore remains outstanding.

**3. [Addressed] Lack of end-to-end system-level evaluation of latency and memory
(by: `jQAv`)**

Reviewer `jQAv` emphasized that FLOPs alone are insufficient to substantiate practical efficiency claims, given that the selector itself introduces additional Transformer layers. The reviewer requested wall-clock latency, throughput, and memory measurements under realistic serving conditions, with explicit inclusion of selector overhead and comparison to training-free baselines.

The authors provided end-to-end latency measurements that include the selector module, and reported memory usage to demonstrate that the proposed approach yields practical inference-time benefits beyond theoretical FLOP reductions.

**4. [Partially addressed] Weak support for video generalization claims
(by: `ChdV`, `jQAv`)**

Reviewer `ChdV` questioned the claim of video generalization, noting that the selector is trained only on images and applied to video frame-by-frame without temporal modeling. Reviewer `jQAv` pointed out that results trail simple baselines on several metrics and that no analysis of temporal coherence, mask stability, or flicker effects was provided.

The rebuttal clarified that video experiments are zero-shot and framed as preliminary, but did not add temporal modeling or deeper analysis. As a result, claims regarding video applicability remain weakly supported.

**5. [Partially addressed] Training stability and justification of Gumbel-Softmax design choices
(by: `t3o2`, `RUPi`)**

Reviewer `t3o2` raised detailed concerns about the training methodology, arguing that the use of Gumbel-Softmax and clipped sparsity penalties appears ad hoc and potentially unnecessary, and questioned whether gradients propagate correctly. Reviewer `RUPi` similarly requested ablations and justification for these design choices.

The authors provided extensive clarifications, code-level explanations, and additional ablation studies comparing Gumbel-based and straight-through estimators. While these responses demonstrate that training is feasible and somewhat stabilize optimization, reviewers remained unconvinced that the chosen formulation is well-motivated or superior in principle. Thus, while clarity improved, concerns about methodological justification remain only partially addressed.

**Reviewer Scores:**

**Reviewer jQAv (4 -> 4)**

jQAv raised central concerns about task–objective misalignment, robustness and worst-case behavior, and the realism of efficiency evaluation. Although end-to-end latency results were added, several requested analyses (e.g., tail-risk robustness, mask stability) remain missing, so no score change is expected.

**Reviewer ChdV (4 -> 4)**

ChdV questioned the theoretical grounding, robustness analysis, and support for video generalization. The rebuttal clarified scope and added limited analysis, but key concerns—particularly around diagnostics and video applicability—remain only partially addressed. The score would likely stay the same.

**Reviewer RUPi (4 -> 4)**

RUPi appreciated the design and OCR results but raised concerns about insufficient analysis depth, unclear baseline reporting, and limited justification for architectural choices (e.g., Gumbel-Softmax and Transformer depth). While the rebuttal added clarifications and some ablations, these concerns were not fully resolved, and the overall assessment would likely remain unchanged.

**Reviewer t3o2 (2 -> 2)**

t3o2 remained unconvinced by the training methodology and design choices, viewing them as insufficiently motivated despite additional explanations. The reviewer’s reject stance would therefore remain unchanged.

---

### Decision · Program_Chairs · 2026-01-26

Reject